# TIMI: Training-Free Image-to-3D Multi-Instance Generation with Spatial Fidelity

Xiao Cai [1]  Pengpeng Zeng [2]  Ji Zhang [3]  Heng Tao Shen [2]  Jingkuan Song [2 4]  Lianli Gao [1]

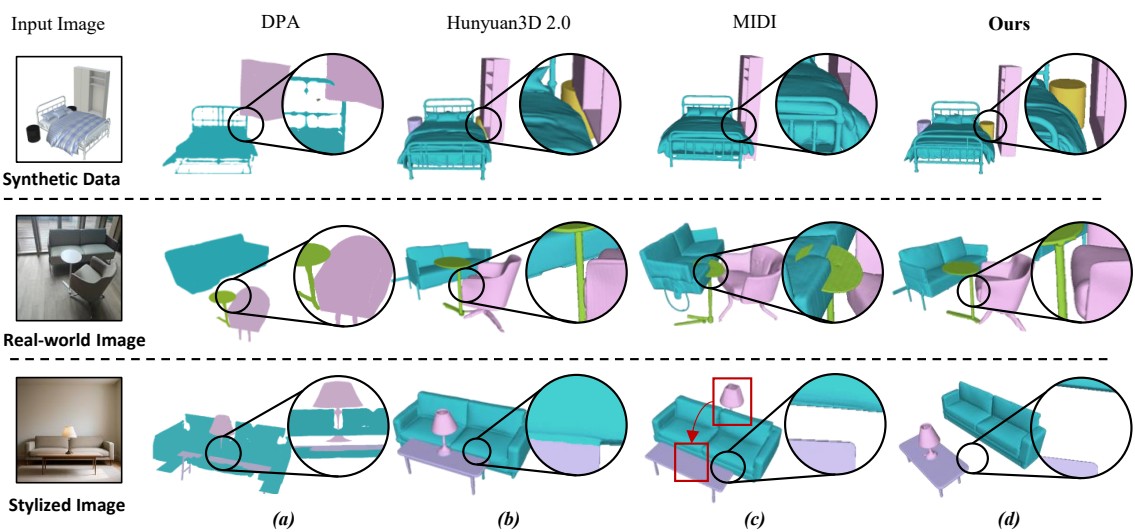

*Figure 1.* **Comparison of our training-free multi-instance 3D generation method with existing approaches.** TIMI generates multiple 3D instances from a single image by training-free guiding pre-trained 3D diffusion models, achieving precise global layout and distinct local instances, without requiring additional fine-tuning.

## Abstract

Precise spatial fidelity in Image-to-3D multi-instance generation is critical for downstream real-world applications. Recent work attempts to address this by fine-tuning pre-trained Image-to-3D (I23D) models on multi-instance datasets, which incurs substantial training overhead and struggles to guarantee spatial fidelity. In fact, we observe that pre-trained I23D models already possess meaningful spatial priors, which remain underutilized as evidenced by instance entanglement issues. Motivated by this, we propose *TIMI*, a novel **T**raining-free framework for **I**mage-to-3D

**M**ulti-**I**nstance generation that achieves high spatial fidelity. Specifically, we first introduce an Instance-aware Separation Guidance (ISG) module, which facilitates instance disentanglement during the early denoising stage. Next, to stabilize the guidance introduced by ISG, we devise a Spatial-stabilized Geometry-adaptive Update (SGU) module that promotes the preservation of the geometric characteristics of instances while maintaining their relative relationships. Extensive experiments demonstrate that our method yields better performance in terms of both global layout and distinct local instances compared to existing multi-instance methods, without requiring additional training and with faster inference speed.

[1]School of Computer Science and Engineering, University of Electronic Science and Technology of China, Chengdu, China [2]School of Computer Science and Technology, Tongji University, Shanghai, China [3]School of Computing and Artificial Intelligence, Southwest Jiaotong University, Chengdu, China [4]Shanghai Innovation Institute, Shanghai, China. Correspondence to: Pengpeng Zeng <is.pengpengzeng@gmail.com>.

*Proceedings of the $43^{rd}$ International Conference on Machine Learning*, Seoul, South Korea. PMLR 306, 2026. Copyright 2026 by the author(s).

## 1. Introduction

Image-to-3D multi-instance (I2MI) generation (Huang et al., 2025; Nie et al., 2020; Zhou et al., 2024) seeks to synthesize 3D scenes containing multiple instances from a single image, serving as a foundation for engineering, product design, and creative industries. In contrast to the rapid progress

in single-instance generation (Liu et al., 2024; Voleti et al., 2024; Shuang et al., 2025; Liang et al., 2024; Wang et al., 2023), I2MI remains challenging and underexplored due to its stringent spatial fidelity requirements, which demand both accurate global layout and distinct local instances.

Recent breakthroughs in diffusion models (Peebles & Xie, 2023) have significantly propelled Image-to-3D (I23D) generation, establishing strong 3D priors that enable high-quality geometry synthesis from a single image. Building upon these advances, early I2MI methods adopt a compositional paradigm (Chen et al., 2024; Zhou et al., 2024; Tang et al., 2025), where individual instances are first generated independently using pre-trained I23D models and then assembled into a shared 3D scene through additional optimization or layout refinement. While straightforward, such multi-stage pipelines are prone to error accumulation, often resulting in inconsistent global layouts, spatial drift, or inter-instance collisions, as illustrated in Fig. 1(a). To overcome these limitations, recent methods such as MIDI (Huang et al., 2025) propose fine-tuning pre-trained I23D models with specialized attention, modeling inter-instance spatial relationships directly within the 3D generation process.

Despite their effectiveness, training-based approaches inevitably introduce substantial training overhead and still fail to fully guarantee spatial fidelity, such as imprecise layouts and fused instances, as shown in Fig. 1(c). Notably, we observe that pre-trained I23D models already possess meaningful spatial priors. As evidenced by Fig. 1(b), Hunyuan3D 2.0 (Team, 2025) can often maintain reasonable inter-instance spatial relationships, but still exhibits instance entanglement. These observations naturally raise the following question: ***Is it possible to exploit and disentangle the spatial priors of pre-trained I23D models for high spatial fidelity I2MI generation in a more flexible manner?***

Inspired by this, we propose ***TIMI***, a novel **T**raining-free framework for **I**mage-to-3D **M**ulti-**I**nstance generation. Specifically, TIMI introduces an Instance-aware Separation Guidance (ISG) module, which performs *instance-aware attention anchoring* aligning early cross-attention with image-derived instance regions, and promotes instance disentanglement through an instance-consistent separation loss. To stabilize this guidance, we further design a Spatial-stabilized Geometry-adaptive Update (SGU) module, which applies *spatial-stabilized regularization* to smooth separation gradients and *geometry-adaptive modulation* to control update magnitudes, thereby helping preserve geometric structure and layout during denoising. Extensive experiments across diverse datasets demonstrate that our method consistently achieves superior global layout accuracy and more distinct local instances (Fig. 1(d)) compared to existing I2MI methods, while requiring no additional training and enabling faster inference (Tab. 1). Our contributions are threefold:

- We propose **TIMI**, a novel **training-free** framework for image-to-3D multi-instance generation with high spatial fidelity.

- We introduce ISG and SGU to jointly promote local instance separation and global layout preservation.

- Extensive experiments demonstrate the superiority of TIMI in both global and local spatial fidelity, achieving state-of-the-art layout alignment and instance distinctiveness without any additional training.

## 2. Related Works

### 2.1. Single-instance 3D Generation

The evolution of single-instance 3D generation reveals a clear transition from optimization-based pipelines to native 3D generation. Early approaches largely relied on Score Distillation Sampling (SDS) (Poole et al., 2023; Lin et al., 2023; Chen et al., 2023) to distill 2D priors into 3D representations, but often suffered from slow optimization and geometric ambiguities. To enhance efficiency and consistency, subsequent research explored multi-view diffusion (Liu et al., 2023; Shi et al., 2023; Yan et al., 2024) and feed-forward reconstruction frameworks (Tang et al., 2024a; Xu et al., 2024; Zou et al., 2024). Most recently, diffusion-based 3D generation has further advanced feed-forward 3D creation by leveraging diffusion models (Rombach et al., 2022; Peebles & Xie, 2023) to directly generate 3D-aware latent representations, enabling high-quality geometry and texture synthesis (Chen et al., 2025b; Cai et al., 2026; Xiang et al., 2025; Team, 2025). These methods achieve impressive fidelity and efficient inference for single-object generation. However, they are primarily designed for isolated object generation and still struggle to generalize to multi-instance scenarios without additional training.

### 2.2. Multi-instance 3D Generation

Multi-instance 3D generation has progressed from traditional reconstruction (Chu et al., 2023; Liu et al., 2022; Paschalidou et al., 2021) and retrieval-based methods (Gao et al., 2024; Gümeli et al., 2022; Kuo et al., 2021) to current generative frameworks. Recent efforts mainly fall into two categories: compositional generation and training-based approaches. Compositional methods (Tang et al., 2024b; Zhou et al., 2024; Rahamim et al., 2024; Chen et al., 2025a) typically follow a "decompose-and-assemble" strategy, where individual instances are generated independently and subsequently composed into a scene using auxiliary cues such as depth, layout, or canonical alignment (Dogaru et al., 2025; Tang et al., 2025; Han et al., 2025; Yao et al., 2025). While flexible, these multi-stage pipelines are often computationally expensive and prone to error accumulation due to the lack of global scene reasoning. In contrast, recent training-

based methods (Huang et al., 2025) pursue an end-to-end solution by fine-tuning I23D models with multi-instance attention to capture inter-instance spatial interactions. Despite their effectiveness, this paradigm incurs substantial training overhead and may weaken learned spatial priors. In light of this, we exploit the strong priors of pre-trained I23D models through training-free instance-aware guidance to enable multi-instance generation with high spatial fidelity.

## 3. Preliminary: 3D Object Generation Models

Our work builds upon large-scale Image-to-3D generation frameworks (Team, 2025; Xiang et al., 2025). These models typically comprise three core components: (1) A 3D Variational Autoencoder (3D VAE) consisting of an encoder $\mathcal{E}$ and a decoder $\mathcal{D}$, which compress 3D geometric representations into a low-dimensional latent space, $z_0 \in \mathbb{R}^{L \times d}$, where $L$ denotes the sequence length and $d$ is the feature dimension. (2) Condition Encoders, which extract semantic feature sequences $c \in \mathbb{R}^{M \times d}$ from the reference image $I$ using pre-trained vision models (e.g., CLIP (Radford et al., 2021) or DINOv2 (Caron et al., 2021)). (3) A Denoising Transformer $\epsilon_\theta$, trained to progressively recover $z_0$ from Gaussian noise $z_T \sim \mathcal{N}(0, \mathbf{I})$.

**Unified Joint Attention Mechanism.** Modern DiT architectures facilitate multi-modal interaction through a joint attention mechanism. Let $z_t$ denote the 3D latent tokens and $c$ denote the image condition tokens. By concatenating them into a joint sequence $U = [c; z_t] \in \mathbb{R}^{(M+L) \times d}$, the interaction is modeled by a global self-attention matrix $\mathbf{A}_{\text{global}}$, which can be decomposed into block-wise components:

$$\mathbf{A}_{\text{global}} = \text{Softmax}\left(\frac{\mathbf{Q}_U \mathbf{K}_U^\top}{\sqrt{d}}\right) = \begin{bmatrix} \mathbf{A}_{cc} & \mathbf{A}_{cz} \\ \mathbf{A}_{zc} & \mathbf{A}_{zz} \end{bmatrix}. \quad (1)$$

Here, the sub-matrix $\mathbf{A}_{zc} \in \mathbb{R}^{L \times M}$ (the bottom-left block) represents the **3D-to-Image Cross-Attention**, governing how 3D geometric tokens query semantic information from the 2D reference, which is therefore the focus of our method.

## 4. Method

In this section, we present a training-free framework *TIMI* for Image-to-3D multi-instance generation, which comprises two components: Instance-aware Separation Guidance (ISG) in Sec. 4.1 and Spatial-stabilized Geometry-adaptive Update (SGU) in Sec. 4.2, as shown in Fig. 2.

### 4.1. Instance-aware Separation Guidance

To alleviate instance entanglement, we introduce the Instance-aware Separation Guidance, which promotes instance-level separation during the early denoising stages.

**Instance-aware Attention Anchoring.** To make instance-level spatial assignment explicit during Image-to-3D denoising, we operate on cross-attention maps in early layers, where 2D–3D correspondences are initially established. In particular, we focus on the 3D-to-image cross-attention matrix $\mathbf{A}_{zc}$, which encodes dense spatial alignments between 3D latent tokens and 2D image features. However, this raw attention remains instance-agnostic and may become ambiguous in multi-instance scenes, especially when different instances overlap spatially in the image.

To address this ambiguity, we introduce an *instance-aware attention anchoring* operation that projects the attention distribution onto discrete semantic instances. Given a set of instance masks $\mathcal{M} = \{M_k\}_{k=1}^K$ extracted from the reference image, we map $\mathbf{A}_{zc}$ to an instance-level representation by computing an *Instance Probability Map* $\mathbf{P} \in \mathbb{R}^{L \times K}$:

$$\mathbf{P}_{\cdot,k} = \mathbf{A}_{zc} \cdot M_k^\top. \quad (2)$$

Here, $\mathbf{P}_{v,k}$ indicates the degree of association between the $v$-th 3D latent token and the $k$-th instance. This projection provides an instance-aligned representation of 3D tokens, serving as the basis for subsequent instance-level separation.

**Instance-consistent Separation Loss.** Based on the instance-aware representation $\mathbf{P}$, our objective is to guide the denoising process toward consistent instance separation in the 3D latent space. Naively encouraging each 3D token to associate with a single instance may lead to degenerate solutions, such as attention collapse around instance centroids, compromising local structure.

To address this issue, we introduce an *instance-consistent separation loss* that incorporates *structure-aware spatial weighting*. Specifically, we derive a spatial weight matrix $\mathbf{W}$ from the masked attention distribution, reflecting the relative spatial support of each token within a given instance:

$$\mathbf{W}_{v,k} = \frac{\mathbf{A}_{zc,v} \cdot M_k}{\sum_{v'}(\mathbf{A}_{zc,v'} \cdot M_k) + \epsilon}. \quad (3)$$

The weight $\mathbf{W}_{v,k}$ emphasizes tokens that are structurally relevant to an instance, allowing the optimization to focus on meaningful spatial regions. Using this instance-aware weighting, we formulate the separation objective as a spatially weighted negative log-likelihood:

$$\mathcal{L}_{\text{sep}} = -\sum_{k=1}^K \sum_{v=1}^L \mathbf{W}_{v,k} \log(\mathbf{P}_{v,k} + \epsilon). \quad (4)$$

This loss promotes separation across instances while encouraging internal consistency within each instance, enabling early and stable instance disentanglement during denoising.

### 4.2. Spatial-stabilized Geometry-adaptive Update

To stabilize instance-aware guidance while preserving coherent spatial structure, we propose the Spatial-stabilized Geometry-adaptive Update.

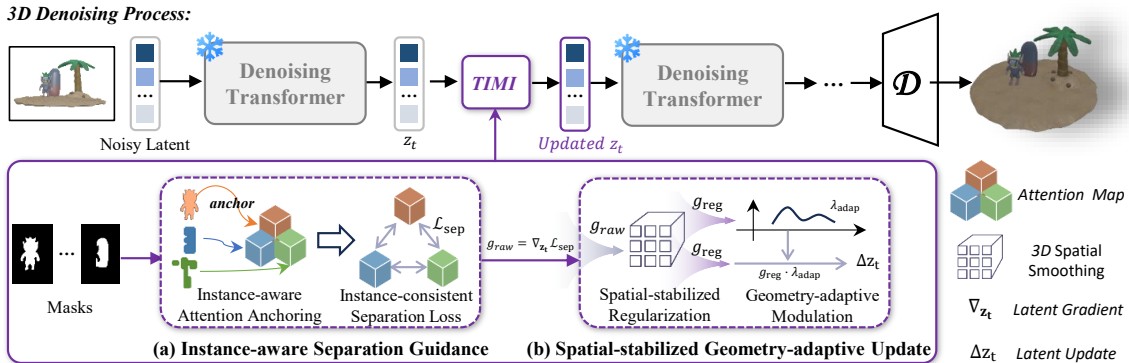

*3D Denoising Process:*

*Figure 2.* **Overview of the TIMI framework.** Given a single image and instance masks, TIMI guides a frozen pre-trained Image-to-3D diffusion model to generate multi-instance 3D outputs without additional training. **(a) Instance-aware Separation Guidance** applies instance-level constraints to early cross-attention layers to promote instance separation. **(b) Spatial-stabilized Geometry-adaptive Update** stabilizes inference-time guidance via geometry-adaptive gradient modulation to preserve overall spatial structure.

**Spatial-stabilized Regularization (SR).** The raw separation gradients $\nabla_{\mathbf{z}}\mathcal{L}_{sep}$ are often sparse and dominated by high-frequency components in the high-dimensional latent space. Directly applying such gradients can disrupt spatial coherence, leading to unstable layouts or fragmented geometric structures. To address this issue, we introduce a spatial-stabilized regularization (SR) on latent features. Specifically, we apply an isotropic 3D Gaussian smoothing to the separation gradients prior to the update:

$$\mathbf{g}_{reg} = \mathcal{K}_{\sigma} * \nabla_{\mathbf{z}}\mathcal{L}_{sep}, \tag{5}$$

where $\mathbf{z} \in \mathbb{R}^{C \times D \times H \times W}$ denotes the 3D latent features, $*$ is the 3D convolution operator, and $\mathcal{K}_{\sigma}$ is a Gaussian kernel with standard deviation $\sigma$. This operation suppresses high-frequency perturbations and enforces spatially coherent gradient propagation, allowing separation guidance to act on contiguous latent regions and helping maintain local geometric continuity during inference-time optimization.

**Geometry-adaptive Modulation (GM).** Different instances exhibit significant variance in geometric sensitivity within the latent space. Applying a uniform update scale may cause excessive deformation for thin or fragile structures, while being insufficient for large or low-response instances. To address this imbalance, we propose a geometry-adaptive gradient modulation strategy based on peak normalization. At each denoising step $t$, we first compute the maximum magnitude of the regularized gradients: $\mu_{max}^{(t)} = \max |\mathbf{g}_{reg}^{(t)}|$. Subsequently, we compute an adaptive scaling factor $\lambda_{adap}$ based on the statistical properties of the current latent feature distribution, ensuring that the maximum feature update is strictly controlled relative to the feature energy scale:

$$\lambda_{adap} = \frac{\alpha \cdot \sigma_{\mathbf{z}_t}}{\mu_{max}^{(t)} + \epsilon}, \tag{6}$$

where $\sigma_{\mathbf{z}_t}$ is the standard deviation of the current latent features $\mathbf{z}_t$, and $\epsilon$ is a stability term ($\epsilon = $ 1e-6) to prevent zero division. Here, $\alpha$ controls the maximum update magnitude relative to the current latent feature scale.

The final update vector $\Delta\mathbf{z}_t$ is then computed as $\Delta\mathbf{z}_t = \lambda_{adap} \cdot \mathbf{g}_{reg}^{(t)}$. By constraining the peak update magnitude, this mechanism balances optimization across heterogeneous geometric structures, preventing collapse in high-gradient regions while maintaining sufficient separation force elsewhere. To further stabilize the optimization trajectory across timesteps, we apply a momentum-based update:

$$\begin{aligned} \mathbf{m}_t &= \beta\mathbf{m}_{t-1} + (1-\beta)\Delta\mathbf{z}_t, \\ \mathbf{z}_t &\leftarrow \mathbf{z}_t - \mathbf{m}_t, \end{aligned} \tag{7}$$

where $\beta$ is the momentum coefficient (set to 0.9), which smooths temporal updates and mitigates oscillations during inference-time optimization.

Overall, the proposed Spatial-stabilized Geometry-adaptive Update transforms raw separation gradients into controlled, geometry-aware latent updates, enabling stable and effective local instance separation while maintaining global layout coherence throughout the denoising process.

## 5. Experiments

### 5.1. Experimental Setup

**Implementation details** We adopt Hunyuan3D 2.0 (Team, 2025) as the base pre-trained I23D model and follow its original inference pipeline and default configurations. Instance masks are extracted using Grounded-SAM (Ren et al., 2024). ISG is applied to early cross-attention layers ($l \leq 4$) during early denoising steps ($t \leq 15$). For SGU, we set the gradient modulation strength to $\alpha = 0.1$ and the Gaussian smoothing standard deviation to $\sigma = 1.5$. We observe that these

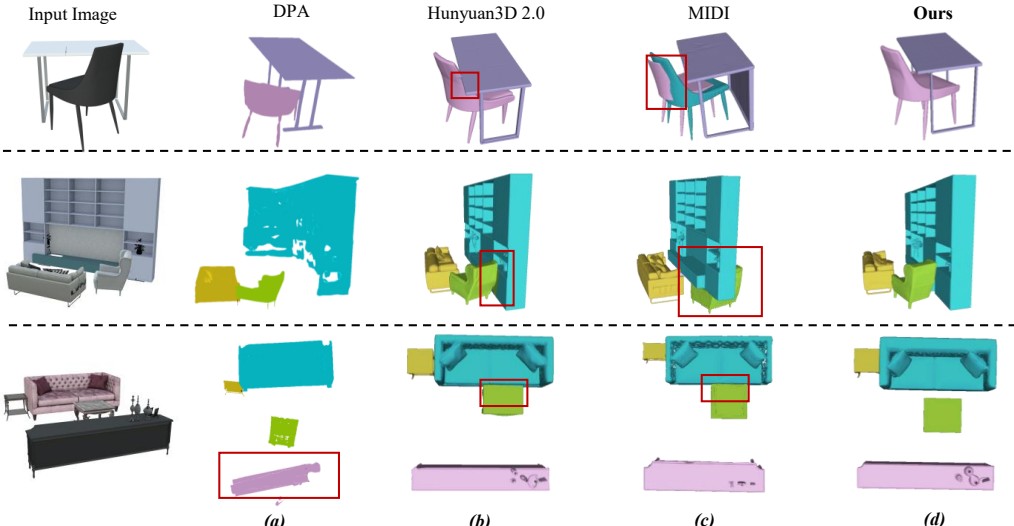

*Figure 3.* **Qualitative Comparisons on Synthetic Data.** Existing methods often produce inaccurate global layouts or fail to separate local instances. Our method preserves both global and local spatial fidelity, yielding well-disentangled instances.

parameters work well across most cases, demonstrating the generalizability of TIMI. We also point out that better results may be obtained with a customized setting, e.g., a larger $\alpha$.

**Datasets.** We evaluate multi-instance 3D generation on three datasets spanning diverse domains. **(i) Synthetic Data:** We randomly sample 30 test scenes from 3D-Front (Fu et al., 2021), each containing two or more spatially adjacent instances. **(ii) Real-world Data:** To assess real-world generalization, we collect 20 images from Real-Data (Quattoni & Torralba, 2009). **(iii) Stylized Data:** To further evaluate cross-domain robustness, we generate 20 stylized multi-instance images using FlUX.1 Kontext (Labs et al., 2025).

**Baselines.** We compare TIMI with representative I2MI approaches. Specifically, we include MIDI (Huang et al., 2025), a training-based method that performs supervised fine-tuning to enable instance disentanglement, and DPA (Zhou et al., 2024), a compositional approach that generates instances independently followed by scene-level composition. In addition, we directly compare against single-instance method Hunyuan3D 2.0 (Team, 2025).

**Metrics.** We evaluate multi-instance 3D generation using complementary metrics that assess both global and local spatial fidelity. Following prior works (Nie et al., 2020; Huang et al., 2025), we report Chamfer Distance (CD) and F-Score (FS) at both global level (CD-S, FS-S) and local level (CD-O, FS-O). To further measure layout alignment and instance disentanglement, we additionally adopt Layout Consistency Distance (LCD) and Separation Success Rate (SSR). Detailed definitions and implementations of the metrics are provided in **Appendix A**.

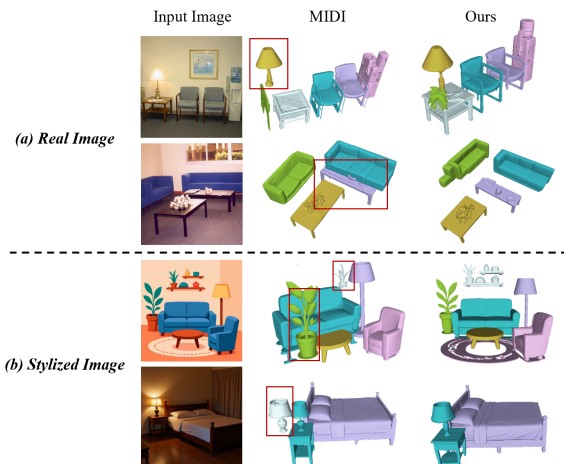

*Figure 4.* **Qualitative comparisons on real and stylized images.** TIMI preserves stronger global and local spatial fidelity across real-world and stylized inputs.

## 5.2. Main Results

### 5.2.1. QUALITATIVE COMPARISON

To qualitatively evaluate the performance of TIMI, we present visual comparisons across diverse datasets.

**I2MI Generation on Synthetic Data.** As illustrated in Fig. 3, TIMI consistently maintains global and local spatial fidelity across synthetic scenes. **(i) Global spatial fidelity.** TIMI faithfully preserves the spatial layout of the input. In contrast, MIDI exhibits noticeable layout drift, exemplified by the misaligned sofa and bookshelf in Fig. 3(c). **(ii) Local spatial fidelity.** At the instance level, TIMI generates struc-

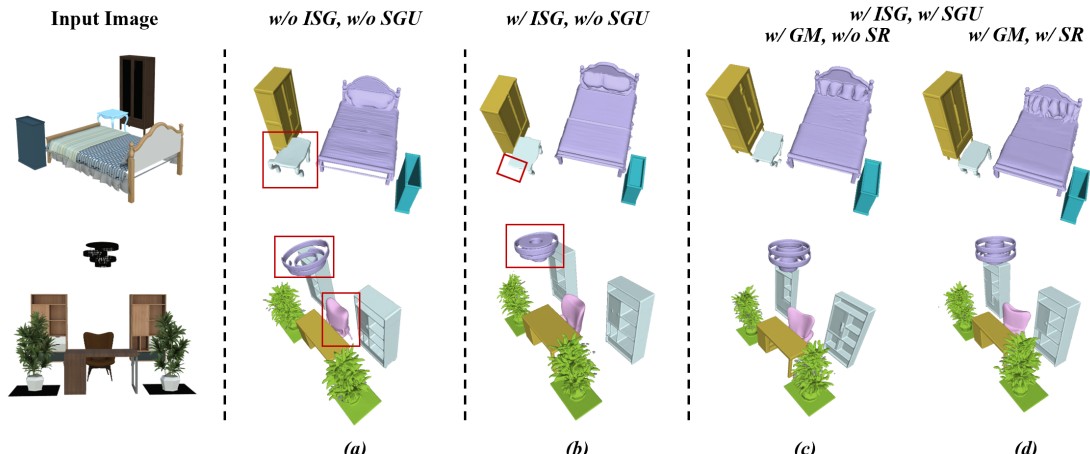

*Figure 5.* **Qualitative ablation study on the effectiveness of proposed components in TIMI.** The results demonstrate that each module progressively enhances the layout alignment and instance distinctiveness, with the full method achieving the best spatial fidelity.

*Table 1.* **Quantitative comparison on synthetic data.** Our method demonstrates superior global and local spatial fidelity while maintaining high inference efficiency.

| Method | Global Spatial Fidelity | | | Local Spatial Fidelity | | | Inference |
|---|---|---|---|---|---|---|---|
| | LCD ↓ | CD-S ↓ | FS-S ↑ | SSR ↑ | CD-O ↓ | FS-O ↑ | Time ↓ |
| Hunyuan3D 2.0 | 0.627 | 0.0492 | 0.450 | 0.697 | 0.0986 | 0.339 | **54.2s** |
| DPA | 0.649 | 0.0662 | 0.249 | 0.743 | 0.1620 | 0.124 | 783s |
| MIDI | 0.634 | **0.0409** | 0.396 | 0.737 | **0.0760** | 0.312 | 90.1s |
| Ours | **0.598** | 0.0424 | **0.458** | **0.809** | 0.0855 | **0.353** | 59.2s |

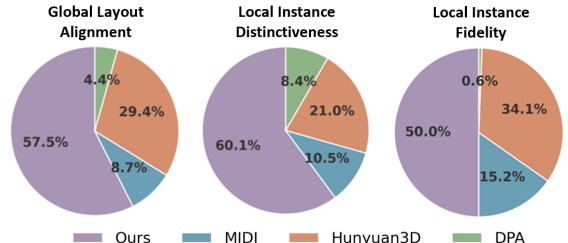

*Figure 6.* **User study.** Our method demonstrates higher subjective preference compared to other methods.

turally distinct instances with crisp boundaries, effectively overcoming the geometric fusion observed in Hunyuan3D 2.0 (e.g., the coalesced sofa and table in Fig. 3(b)). Collectively, these results confirm TIMI's capacity to maintain precise global layouts and distinct local instances.

**I2MI Generation from Real and Stylized Images.** We further extend our evaluation to real-world and stylized domains (Fig. 4), where TIMI continues to uphold high spatial fidelity. Specifically, in terms of global layout, MIDI exhibits noticeable spatial drift, where instances such as plants are incorrectly placed in front of the sofa and table in Fig. 4(b), while our results remain well aligned with the input images. Regarding local instance, MIDI suffers from instance fusion, as evidenced by the overlapping table and sofa in Fig. 4(a), whereas TIMI preserves clear instance separation. These observations validate the effective generalization of our framework under varied inputs.

### 5.2.2. QUANTITATIVE COMPARISON

To systematically evaluate TIMI, we conduct quantitative comparisons with existing methods, as summarized in Tab. 1. The results reveal that: **(i)** Despite being training-free, TIMI consistently outperforms the training-based approach MIDI in both global and local spatial fidelity. **(ii)** TIMI achieves the strongest overall global spatial fidelity, attaining the

best LCD (**0.598**) and FS-S (**0.458**). Although MIDI yields a slightly lower CD-S, LCD more directly reflects layout alignment, indicating that TIMI reconstructs global layouts that are better aligned with the input image. **(iii)** For local spatial fidelity, TIMI substantially surpasses all baselines, achieving the highest SSR (**0.809**) and FS-O (**0.353**), demonstrating superior instance-level separation. While MIDI exhibits competitive CD-O, its lower SSR suggests more frequent instance fusion, whereas TIMI generates more distinct and well-separated instances. **(iv)** In addition, TIMI maintains high inference efficiency (∼59s), comparable to the base model Hunyuan3D, and is significantly faster than both the training-based MIDI (∼90s) and the compositional method DPA (∼783s). The quantitative evidence verifies that TIMI enhances both layout alignment and instance separation without sacrificing the efficiency of generation.

**User Study.** We further conduct a user study to assess perceptual quality under human evaluation. As summarized in Fig. 6, **(i)** 57.5% of participants prefer our results in terms of global layout alignment. **(ii)** 60.1% of users favor our method for local instance distinctiveness, significantly outperforming MIDI and Hunyuan3D, suggesting that fine-

*Table 2.* **Quantitative ablation study on components in TIMI,** progressively incorporating ISG and SGU effectively enhances the global and local spatial fidelity.

| | Global Spatial Fidelity | | | Local Spatial Fidelity | | |
|---|---|---|---|---|---|---|
| | LCD ↓ | CD-S ↓ | FS-S ↑ | SSR ↑ | CD-O ↓ | FS-O ↑ |
| w/o ISG, w/o SGU | 0.627 | 0.0492 | 0.450 | 0.697 | 0.0986 | 0.339 |
| w/ ISG, w/o SGU | 0.623 | 0.0462 | 0.448 | 0.674 | 0.0864 | 0.351 |
| w/ ISG, w/ SGU | **0.598** | **0.0424** | **0.458** | **0.809** | 0.0855 | **0.353** |

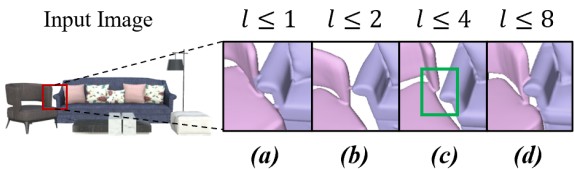

*Figure 7.* **Qualitative results of ablation study on $l$ in ISG.** $l \leq 4$ achieves the best instance separation, while smaller or larger $l$ leads to insufficient disentanglement or partial instance fusion.

*Table 3.* **Quantitative results of ablation study on $l$ in ISG.**

| $l$ | Global Spatial Fidelity | | | Local Spatial Fidelity | | |
|---|---|---|---|---|---|---|
| | LCD ↓ | CD-S ↓ | FS-S ↑ | SSR ↑ | CD-O ↓ | FS-O ↑ |
| 1 | 0.600 | 0.0437 | 0.454 | 0.772 | 0.0896 | **0.372** |
| 2 | 0.615 | 0.0443 | 0.453 | 0.785 | 0.0874 | 0.352 |
| 3 | **0.554** | 0.0449 | 0.439 | 0.797 | 0.1007 | 0.360 |
| 4 | 0.598 | **0.0424** | **0.457** | **0.809** | **0.0855** | 0.353 |
| 5 | 0.594 | 0.0434 | 0.452 | 0.786 | 0.0938 | 0.359 |
| 8 | 0.616 | 0.0468 | 0.442 | 0.773 | 0.0931 | 0.358 |

grained instance separation is more perceptually evident to human observers. **(iii)** Our method is also preferred by 50.0% of users in terms of instance quality, indicating that improved global and local spatial fidelity is achieved without compromising generative quality.

## 5.3. Ablation Study

In this section, we conduct comprehensive ablation studies to validate the effectiveness of components within our TIMI. First, we verify the necessity of the two core modules: ISG and SGU. Next, we provide a fine-grained analysis within ISG in Sec. 5.3.1 and SGU in Sec. 5.3.2.

**Effect of *ISG* and *SGU*.** In this experiment, we investigate the effectiveness of ISG and SGU by progressively adding them to the baseline. From the results in Tab. 2, we have the following observations. **(i)** Both ISG and SGU contribute to the improvement of global and local spatial fidelity. **(ii)** *w/ ISG.* Introducing the ISG effectively improves local spatial fidelity, as reflected by the gains in FS-O and CD-O. **(iii)** *w/ ISG, w/ SGU.* Further incorporating SGU substantially enhances global spatial fidelity while preserving the improved local instance quality, resulting in consistent gains across global and local metrics.

These findings are further supported by qualitative comparisons in Fig. 5. **(i)** Without either module (Fig. 5(a)), instances such as the stool and cabinet exhibit severe fusion with the stool appearing incomplete and deformed. **(ii)** With ISG enabled via naive gradient updates (Fig. 5(b)), the stool becomes disentangled but suffers from geometric fractures (e.g., missing legs), confirming that direct gradient application disrupts 3D structural coherence. **(iii)** By incorporating both ISG and SGU (Fig. 5(d)), the overall layout becomes more coherent, and the stool is correctly generated with complete structure and plausible geometry.

### 5.3.1. ANALYSIS OF ISG

To further analyze the design choices of ISG, we conduct ablation studies on two key hyperparameters: (i) the selection of guided cross-attention layers $l$, and (ii) the number of early denoising timesteps $t$ over which ISG is applied.

**Effect of $l$ in ISG.** Tab. 3 reports the performance of ISG when applying guidance to different ranges of cross-attention layers. We summarize three key observations. **(i)** Guiding the first four layers ($l \in [0, 4]$) achieves the best overall trade-off, delivering strong global spatial fidelity (lowest CD-S), and the most effective instance disentanglement (highest SSR). **(ii)** Restricting guidance to very shallow layers (e.g., $l \in [0, 1]$) slightly improves FS-related metrics but results in degraded global spatial fidelity and weaker instance separation, indicating insufficient spatial regulation. **(iii)** Increasing the layer coverage beyond the fourth layer (e.g., up to layer 8) brings no further improvement and instead results in a consistent performance drop compared to the four-layer setting. This indicates that deeper layers may capture higher-level semantics (e.g., texture), which are less effective for instance-level spatial regulation.

Fig. 7 provides qualitative comparisons under different choices of guided cross-attention layers $l$. **(i)** When guidance is applied to the first four layers ($l \in [0, 4]$), instance separation is the most visually clear and stable. As shown, the sofa armrest and the chair are cleanly disentangled with well-defined boundaries. **(ii)** For shallower guidance ranges (e.g., $l \in [0, 1]$ or $[0, 2]$), noticeable instance entanglement is observed. In particular, structural parts of the sofa tend to merge with the nearby chair, indicating that guidance confined to very early layers is insufficient. **(iii)** Extending guidance to deeper layers (e.g., $l \in [0, 8]$) also degrades separation quality. Although the overall geometry remains plausible, object boundaries become less distinct, and partial fusion reappears between adjacent instances. This suggests that deeper layers, which may encode more abstract or semantic information, are less suitable for enforcing instance-level spatial separation. Overall, the qualitative

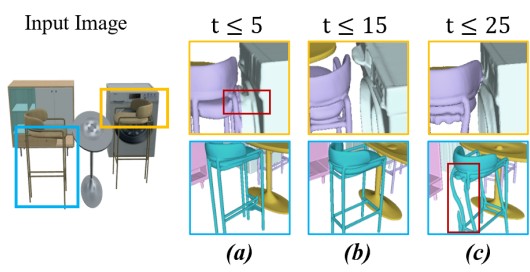

*Figure 8.* **Qualitative ablation study on $t$ in ISG.** $t \leq 15$ yields the best trade-off, while shorter or longer schedules cause insufficient disentanglement or geometric distortion.

*Table 4.* **Quantitative results of ablation study on $t$ in ISG.**

| $t$ | Global Spatial Fidelity | | | Local Spatial Fidelity | | |
|---|---|---|---|---|---|---|
| | LCD $\downarrow$ | CD-S $\downarrow$ | FS-S $\uparrow$ | SSR $\uparrow$ | CD-O $\downarrow$ | FS-O $\uparrow$ |
| 5 | 0.554 | 0.0448 | 0.446 | 0.753 | 0.1041 | 0.353 |
| 10 | 0.563 | 0.0435 | 0.453 | 0.739 | 0.0859 | **0.383** |
| 15 | 0.598 | **0.0424** | **0.458** | **0.809** | **0.0855** | 0.353 |
| 20 | 0.603 | 0.0443 | 0.437 | 0.790 | 0.0991 | 0.335 |
| 25 | **0.552** | 0.0433 | 0.442 | 0.771 | 0.0976 | 0.342 |

results confirm that guiding the first four layers achieves the best balance between instance disentanglement and structural preservation, in line with the quantitative performance.

**Effect of $t$ in ISG.** We further analyze the effect of applying ISG at different denoising timesteps, with quantitative results summarized in Tab. 4. **(i)** Applying ISG within the first 15 denoising steps achieves the most balanced performance, yielding the best global layout consistency (lowest CD-S) and the highest local instance distinctiveness (SSR), while maintaining competitive instance quality. **(ii)** Limiting guidance to an overly short early duration ($t \leq 10$) constrains instance disentanglement, as indicated by reduced SSR. **(iii)** Conversely, extending ISG into later denoising stages ($t \geq 20$) brings no further gains in spatial fidelity and instead degrades instance separation (lower FS-O).

These trends are visually supported in Fig. 8. Applying ISG only at very early steps ($t \leq 5$) results in insufficient separation, with the chair and washing machine remaining fused (Fig. 8(a)), whereas overly prolonged guidance ($t \leq 25$) introduces structural distortions, such as deformed chair legs (Fig. 8(c)). Therefore, we select $t = 15$ as the default setting, which provides a favorable balance between global layout alignment and local instance distinctiveness.

### 5.3.2. ANALYSIS OF SGU

We conduct a detailed analysis of the SGU. Specifically, we first evaluate the individual contributions of its components, and then examine the effects of the guidance strength $\alpha$ in Eq. 6 and the spatial regularization factor $\sigma$ in Eq. 5.

*Table 5.* **Quantitative ablation study on components in SGU,** showing that GM and SR jointly enhance global and local spatial fidelity.

| | Global Spatial Fidelity | | | Local Spatial Fidelity | | |
|---|---|---|---|---|---|---|
| | LCD $\downarrow$ | CD-S $\downarrow$ | FS-S $\uparrow$ | SSR $\uparrow$ | CD-O $\downarrow$ | FS-O $\uparrow$ |
| w/o GM, w/o SR | 0.623 | 0.0462 | 0.448 | 0.674 | 0.0864 | 0.351 |
| w/ GM, w/o SR | **0.597** | 0.0455 | 0.444 | 0.764 | 0.0882 | **0.368** |
| w/ GM, w/ SR | 0.598 | **0.0424** | **0.458** | **0.809** | 0.0855 | 0.353 |

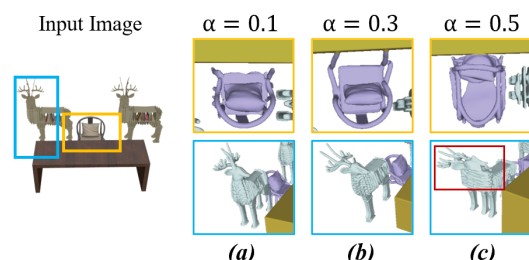

*Figure 9.* **Qualitative results of ablation study on $\alpha$ in SGU.** A moderate guidance strength ($\alpha = 0.1$) yields the most balanced global layout consistency and instance-level geometric fidelity.

**Effect of *GM* and *SR* in SGU.** Tab. 5 reports the ablation results of the individual components in SGU. **(i)** Without GM and SR, the model exhibits inferior global spatial fidelity and weak instance disentanglement, as reflected by higher CD-S and the lowest SSR. **(ii)** Introducing GM alone substantially improves global layout consistency (best LCD), but provides limited gains in instance disentanglement and local spatial fidelity. **(iii)** Combining GM with SR yields consistent improvements across all metrics, achieving the best global and local spatial fidelity.

As shown in Fig. 5, with GM introduced into SGU, the model achieves a more stable and coherent global layout. However, noticeable instance-level structural distortions emerge, such as the twisted stool legs in Fig. 5(c). After further incorporating SR, the global layout remains stable while such distortions are effectively alleviated, enabling robust local instance separation.

**Effect of $\alpha$ in SGU.** We investigate the sensitivity of the global guidance strength $\alpha$, with detailed results provided in Tab. 6. **(i)** When $\alpha$ is set to a small value (e.g., $\alpha = 0.1$), SGU generally achieves strong global spatial fidelity while preserving robust local instance quality. **(ii)** Slightly increasing $\alpha$ (e.g., $\alpha = 0.2$) maintains comparable global spatial fidelity, but local instance quality begins to degrade, as reflected by increased instance-level CD. **(iii)** As $\alpha$ is further increased to intermediate ranges (e.g., 0.3 or 0.4), both global and local spatial fidelity consistently deteriorate. **(iv)** For large values of $\alpha$ (e.g., $\alpha = 0.5$), although global alignment may remain strong, excessive guidance tends to destabilize instance geometry (degraded FS metrics).

*Table 6.* **Quantitative results of ablation study on $\alpha$ in SGU.**

| $\alpha$ | Global Spatial Fidelity | | | Local Spatial Fidelity | | |
|---|---|---|---|---|---|---|
| | LCD ↓ | CD-S ↓ | FS-S ↑ | SSR ↑ | CD-O ↓ | FS-O ↑ |
| 0.1 | 0.598 | 0.0424 | **0.458** | **0.809** | 0.0855 | 0.353 |
| 0.2 | **0.523** | 0.0439 | 0.446 | 0.780 | 0.1001 | **0.359** |
| 0.3 | 0.562 | 0.0449 | 0.441 | 0.796 | 0.0897 | 0.326 |
| 0.4 | 0.611 | 0.0428 | 0.436 | 0.766 | 0.0862 | 0.315 |
| 0.5 | 0.570 | **0.0414** | 0.429 | 0.750 | **0.0755** | 0.314 |

These trends are further illustrated in Fig. 9. With smaller $\alpha$, instances exhibit more complete structures and clearer boundaries, whereas larger $\alpha$ values progressively introduce geometric degradation and instance interference, indicating that overly strong guidance can disrupt geometric stability and compromise spatial fidelity.

**Effect of $\sigma$ in SGU.** We analyze the influence of $\sigma$ in Eq. 5, with quantitative results reported in Tab. 7. We observe that a small $\sigma$ (e.g., $0.5$) leads to inferior global spatial fidelity and weaker instance-level separation, indicating insufficient spatial regularization. Conversely, an overly large $\sigma$ (e.g., $2.5$) degrades both global and local spatial fidelity, suggesting that overly strong smoothing may disrupt instance geometry. Consequently, we adopt $\sigma = 1.5$ as the optimal setting, which achieves the best balance between layout consistency and instance distinctiveness (best CD-S and SSR).

We further conduct a qualitative comparison to investigate the impact of different $\sigma$ values, as visualized in Fig. 10. As shown in Fig. 10(a), an overly small $\sigma$ leads to unstable guidance, resulting in geometric fractures (e.g., the chair back is detached from its body) and insufficient disentanglement (e.g., the chair remains fused with the bed). This indicates that weak smoothing fails to enforce coherent separation forces. Conversely, as shown in Fig. 10(c), while an overly large $\sigma$ provides sufficient smoothing to separate the chair from the bed, it disrupts the intrinsic geometry of individual instances, causing noticeable structural distortions such as the twisted furniture leg. Based on these observations, we adopt $\sigma = 1.5$ as the optimal default setting to balance separation effectiveness and geometric preservation.

## 6. Conclusion

We introduce TIMI, a training-free Image-to-3D multi-instance generation framework via instance-aware guidance. By exploiting the spatial priors of pre-trained I23D models, TIMI improves multi-instance generation without additional fine-tuning. The proposed ISG promotes local instance disentanglement during early denoising, while SGU stabilizes the guidance to preserve geometric structures and precise global layouts. Experiments across synthetic, real-world,

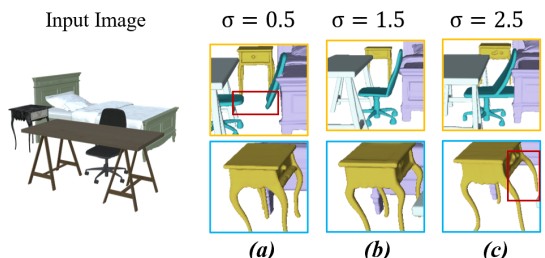

Input Image    $\sigma = 0.5$    $\sigma = 1.5$    $\sigma = 2.5$

*(a)*    *(b)*    *(c)*

*Figure 10.* **Qualitative results of ablation study on $\sigma$ in SGU.** $\sigma = 1.5$ achieves the best balance between instance separation and geometric preservation, while smaller or larger $\sigma$ leads to insufficient disentanglement or structural distortion.

*Table 7.* **Quantitative results of ablation study on $\sigma$ in SGU.**

| $\sigma$ | Global Spatial Fidelity | | | Local Spatial Fidelity | | |
|---|---|---|---|---|---|---|
| | LCD ↓ | CD-S ↓ | FS-S ↑ | SSR ↑ | CD-O ↓ | FS-O ↑ |
| 0.5 | **0.556** | 0.0439 | 0.444 | 0.782 | 0.1057 | 0.352 |
| 1.0 | 0.622 | 0.0442 | 0.448 | 0.768 | **0.0837** | **0.379** |
| 1.5 | 0.598 | **0.0424** | **0.458** | **0.809** | 0.0855 | 0.353 |
| 2.0 | 0.570 | 0.0428 | 0.444 | 0.784 | 0.0942 | 0.340 |
| 2.5 | 0.638 | 0.0436 | 0.440 | 0.782 | 0.1032 | 0.332 |

and stylized inputs demonstrate that TIMI achieves superior global and local spatial fidelity over existing methods while maintaining efficient inference. These results suggest that TIMI offers a practical solution for high spatial fidelity multi-instance 3D generation.

## Acknowledgments

This study is supported by grants from Fundamental and Interdisciplinary Disciplines Breakthrough Plan of the Ministry of Education of China (No. JYB2025XDXM116), and the National Natural Science Foundation of China (Grant No. 62425208, No. U22A2097, No. U23A20315, No. 82441006).

## Impact Statement

This paper presents TIMI, a framework designed to advance 3D generative modeling by enabling high spatial fidelity multi-instance synthesis without the substantial computational costs of fine-tuning. By introducing a training-free paradigm, our work lowers the barrier to entry for high-quality 3D content creation, potentially democratizing access for applications in industrial design, virtual reality, and the creative industries. However, while our method significantly improves efficiency, it relies on pre-trained foundation models and may inherently reflect their underlying data biases. We do not foresee immediate negative societal consequences beyond general considerations regarding the responsible use of generative AI technologies.

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

## A. Experimental Setup

**Metrics.** We employ multi-dimensional evaluation metrics to assess multi-instance generation, focusing on both global and local spatial fidelity.

**(i) For Objective Evaluation**, following prior works (Nie et al., 2020; Huang et al., 2025), we adopt Chamfer Distance (CD) and F-Score (FS), which provide a general measure of reconstruction quality and spatial fidelity at the global scene level (CD-S, FS-S) and local object level (CD-O, FS-O). To further capture global layout alignment and instance-level separation, we introduce two complementary metrics.

Layout Consistency Distance (LCD) evaluates the accuracy of object placement by computing the centroid-based Chamfer Distance between predicted and ground-truth instances:

$$\text{LCD} = \frac{1}{|\mathcal{C}_{pred}|} \sum_{x \in \mathcal{C}_{pred}} \min_{y \in \mathcal{C}_{gt}} \|x - y\|_2 \qquad (8)$$

$$+ \frac{1}{|\mathcal{C}_{gt}|} \sum_{y \in \mathcal{C}_{gt}} \min_{x \in \mathcal{C}_{pred}} \|y - x\|_2, \qquad (9)$$

where $\mathcal{C}_{pred}$ and $\mathcal{C}_{gt}$ denote the sets of centroids of predicted and ground-truth object instances, respectively, and $\|\cdot\|_2$ represents the Euclidean distance between two points. The first term measures how close each predicted object is to its nearest ground-truth object, while the second term ensures all ground-truth instances are covered by predictions.

Separation Success Rate (SSR) measures how well the predicted number of independent instances matches the ground truth, reflecting instance distinctiveness regardless of exact shape or position:

$$\text{SSR} = \frac{\min(N_{pred}, N_{gt})}{\max(N_{pred}, N_{gt})}, \qquad (10)$$

where $N_{pred}$ and $N_{gt}$ denote the number of predicted and ground-truth independent instances, respectively.

**(ii) For Subjective Evaluation**, we conduct a user study where participants perform blind comparative ratings on randomly paired results based on three criteria: Global Layout Alignment, Local Instance Distinctiveness, and Local Instance Fidelity.

## B. More Experimental Results

### B.1. Large-Scale Quantitative Evaluation

Table 8 reports the quantitative results on a larger test set of 500 samples. **(i)** Compared with the base Hunyuan3D 2.0 model, TIMI consistently improves both global spatial fidelity and local instance quality. In particular, TIMI achieves lower LCD and CD-S, indicating more accurate scene-level layout and geometry, while also improving FS-S.

*Table 8.* Large-scale quantitative evaluation results. $\pm$ denotes the 95% confidence interval.

| Method | LCD ↓ | CD-S ↓ | FS-S ↑ |
|---|---|---|---|
| DPA | 0.634±0.020 | 0.0711±0.0342 | 0.193±0.013 |
| Hunyuan3D 2.0 | 0.609±0.023 | 0.0451±0.0029 | 0.398±0.018 |
| MIDI | 0.608±0.023 | 0.0494±0.0031 | 0.324±0.017 |
| **TIMI** | **0.596±0.023** | **0.0426±0.0029** | **0.410±0.018** |

| Method | SSR ↑ | CD-O ↓ | FS-O ↑ |
|---|---|---|---|
| DPA | 0.652±0.021 | 0.1215±0.0060 | 0.120±0.010 |
| Hunyuan3D 2.0 | 0.695±0.019 | 0.1062±0.0071 | 0.268±0.018 |
| MIDI | 0.690±0.018 | 0.1129±0.0062 | 0.189±0.014 |
| **TIMI** | **0.705±0.020** | **0.1024±0.0069** | **0.274±0.018** |

**(ii)** Compared with MIDI, TIMI obtains better performance across all reported metrics under this larger-scale setting. This suggests that the improvement is not limited to a small set of examples, but remains stable across a more diverse evaluation set. **(iii)** The overall trends are consistent under the 95% confidence intervals, providing stronger evidence that TIMI improves multi-instance spatial fidelity in a reliable manner.

*Table 9.* Cross-backbone generalization results on Trellis.

| Method | LCD ↓ | CD-S ↓ | FS-S ↑ | SSR ↑ | CD-O ↓ | FS-O ↑ |
|---|---|---|---|---|---|---|
| Trellis | 0.692 | 0.0668 | 0.221 | 0.776 | 0.0899 | 0.131 |
| Trellis + TIMI | **0.635** | **0.0622** | **0.273** | **0.785** | **0.0813** | **0.156** |

### B.2. Cross-Backbone Generalization

Table 9 evaluates the generalizability of TIMI by applying it to the Trellis backbone without modifying the model architecture or performing additional training. **(i)** TIMI consistently improves Trellis on all global metrics, reducing LCD from 0.692 to 0.635 and CD-S from 0.0668 to 0.0622, while increasing FS-S from 0.221 to 0.273. This indicates that TIMI can improve scene-level layout and geometry over the original backbone. **(ii)** TIMI also improves all local instance-level metrics, including SSR, CD-O, and FS-O, suggesting better instance separation and local object fidelity. **(iii)** Although the absolute performance still depends on the capacity of the underlying backbone, the consistent improvements on Trellis demonstrate that TIMI is not restricted to Hunyuan3D 2.0. Instead, it acts as a general training-free guidance mechanism that can be plugged into different I23D backbones.

