# OpenReview forum: "TIMI: Training-Free Image-to-3D Multi-Instance Generation with Spatial Fidelity"
_ICML.cc/2026/Conference — ICML 2026 regular_

### Official Review · Reviewer_92rs · 2026-02-28

**Soundness:** 3
**Presentation:** 3
**Significance:** 3
**Originality:** 3
**Overall Recommendation:** 4
**Confidence:** 4

**Summary:**

This paper proposes TIMI, a training-free framework for single-image multi-instance Image-to-3D generation that improves spatial fidelity by intervening in cross-attention during early denoising (ISG) and stabilizing geometry via SGU. The method aims to leverage spatial priors already present in pretrained I2-3D models to avoid costly multi-instance fine-tuning.

**Compliance With Llm Reviewing Policy:**

Affirmed.

**Final Justification:**

The paper presents a practical and reasonably sound training-free approach for multi-instance image-to-3D generation, with strengths in motivation, technical design, and potential practical significance. My main concerns were the small evaluation scale, missing cross-backbone evidence, and unclear robustness to noisy masks. The rebuttal usefully addressed these by adding a 500-sample evaluation with confidence intervals, validation on Trellis, and mask perturbation experiments covering boundary noise, dropout, and missing instances. As a result, most of my concerns were addressed, which reinforces my prior view of the paper.

I therefore keep my original score.

**Key Questions For Authors:**

1. Scalability and statistical validity. Can the authors provide results on substantially larger and more diverse real-image benchmarks, along with variance or confidence intervals for all quantitative metrics? This would materially affect my confidence in the reported gains.

2. Cross-model generalization. How does TIMI perform when applied to other I2-3D backbones with different architectures (e.g., Trellis, sam3d)? Demonstrating consistent improvements would significantly strengthen the “training-free framework” claim.

3. Mask robustness. How sensitive is the method to degraded or noisy instance masks? Please provide controlled experiments where mask quality is systematically perturbed (e.g., IoU sweeps, missing instances, boundary noise).

**Limitations:**

No. The current limitation discussion does not sufficiently address the strong dependence on mask quality or the lack of cross-backbone validation. The authors should explicitly discuss these risks and the conditions under which TIMI may fail.

**Strengths And Weaknesses:**

**Strengths**

The paper identifies an important and practically relevant problem: instance entanglement in single-image multi-instance 3D generation. The core observation that pretrained I2-3D models already encode useful spatial priors and can be steered at inference time is interesting and potentially impactful for reducing training overhead. The proposed ISG and SGU pipeline is conceptually coherent, and qualitative results suggest improved instance separation and layout consistency over several baselines. The training-free and plug-in nature of the method is highly appealing from an engineering and deployment perspective.

**Weaknesses**

Despite these merits, the submission suffers from an insufficient scale and statistical rigor in its experimental evaluation. The paper evaluates only 30 synthetic scenes, 20 real images, and 20 stylized images while making broad claims about spatial fidelity and generalization. For a venue like ICML, this evaluation is severely underpowered. The absence of variance, confidence intervals, or statistical significance for metrics such as LCD and SSR makes it difficult to determine whether the reported improvements are robust or merely artifacts of sample noise. Consequently, the current evidence resembles a small-scale case study rather than a reliable benchmark. This limited evaluation is compounded by a lack of cross-backbone validation. Although TIMI is positioned as a general training-free framework, all experiments rely exclusively on Hunyuan3D-2.0. Because the method intervenes directly in cross-attention, its effectiveness may be highly architecture-dependent. Without demonstrating consistent gains on at least one structurally different I2-3D model, it is unclear if TIMI is a broadly applicable framework or a model-specific engineering patch. Furthermore, the proposed method exhibits a strong and largely unexamined dependence on high-quality 2D instance masks. The ISG module relies critically on the instance probability map $P$, derived from Grounded-SAM masks, yet the authors do not study robustness to realistic segmentation noise such as boundary errors, missed instances, or occlusion. Since mask quality is rarely ideal in practical pipelines, this lack of robustness analysis raises significant concerns about the stability and real-world reliability of the guidance mechanism.

---

> ### Author Rebuttal · Authors · 2026-03-31
>
> We thank the reviewer for acknowledging the contributions of this work and insightful suggestions.  We hope the following clarifications are helpful in addressing the concerns.
>
> **W1/Q1: Scalability and statistical validity**
>
> **Response:** To address concerns about statistical confidence, we expand the test set to 500 samples and report the updated results in the table, providing a substantially larger and more representative evaluation setting. We observe that the overall trends remain consistent: TIMI achieves the best performance in terms of spatial fidelity across metrics under the larger evaluation setting. We further report 95% confidence intervals for all metrics, which provide a clearer view of variation across samples. These results indicate that ***the observed improvements are not driven by small-sample effects, but remain stable across a more diverse dataset***, providing stronger empirical support for the robustness and reliability of our conclusions.
>
> | Method        | LCD ↓             | CD_S ↓              | FS_S ↑            | SSR ↑             | CD_O ↓              | FS_O ↑            |
> | ------------- | ----------------- | ------------------- | ----------------- | ----------------- | ------------------- | ----------------- |
> | Hunyuan3D 2.0 | 0.609 ± 0.023     | 0.0451 ± 0.0029     | 0.398 ± 0.018     | 0.695 ± 0.019     | 0.1062 ± 0.0071     | 0.268 ± 0.018     |
> | MIDI          | 0.608 ± 0.023     | 0.0494 ± 0.0031     | 0.324 ± 0.017     | 0.690 ± 0.018     | 0.1129 ± 0.0062     | 0.189 ± 0.014     |
> | TIMI          | **0.596 ± 0.023** | **0.0426 ± 0.0029** | **0.410 ± 0.018** | **0.705 ± 0.020** | **0.1024 ± 0.0069** | **0.274 ± 0.018** |
>
> **W2/Q2: Cross-model generalization**
>
> **Response:** To validate the generalizability of TIMI across different I23D backbones, we further apply TIMI to another representative model, Trellis[1], without modifying its architecture. The qualitative results are shown in R-Fig.3 (***https://anonymous.4open.science/r/TIMI-rebuttal-7BD9***), and the quantitative results are reported below. We observe consistent improvements across both global and local spatial fidelity on Trellis. These results indicate that ***TIMI is not tied to a specific architecture and can generalize across different I23D models***, supporting its formulation as a general training-free framework.
>
> | Method         | LCD ↓     | CD_S ↓     | FS_S ↑    | SSR ↑     | CD_O ↓     | FS_O ↑    |
> | -------------- | --------- | ---------- | --------- | --------- | ---------- | --------- |
> | Trellis        | 0.692     | 0.0668     | 0.221     | 0.776     | 0.0899     | 0.131     |
> | Trellis + TIMI | **0.635** | **0.0622** | **0.273** | **0.785** | **0.0813** | **0.156** |
>
> **W3/Q3: Mask robustness**
>
> **Response:** We agree that robustness to imperfect masks is important in practical settings. In practice, masks produced by Grounded-SAM already contain various imperfections (e.g., boundary noise, overlap, and missing regions), as illustrated in R-Fig.1 (***https://anonymous.4open.science/r/TIMI-rebuttal-7BD9***), while TIMI still produces spatially coherent 3D results.
>
> To further validate this, we conduct controlled experiments by systematically perturbing the input masks, including (1) boundary noise via erosion/dilation, (2) missing regions via random dropout, and (3) instance-level failures such as missing instances, with varying levels of degradation. The results below show that ***TIMI remains stable under moderate perturbations***, with only gradual performance degradation as mask quality decreases. This suggests that TIMI can operate reliably with imperfect masks in practical scenarios.
>
> | Mask Setting     | LCD ↓ | CD-S ↓ | FS-S ↑ | SSR ↑ | CD-O ↓ | FS-O ↑ |
> | ---------------- | ----- | ------ | ------ | ----- | ------ | ------ |
> | Clean            | 0.551 | 0.0468 | 0.428  | 0.672 | 0.0894 | 0.336  |
> | Boundary Noise   | 0.559 | 0.0553 | 0.414  | 0.728 | 0.1087 | 0.315  |
> | Dropout          | 0.630 | 0.0525 | 0.408  | 0.662 | 0.1032 | 0.308  |
> | Missing Instance | 0.572 | 0.0521 | 0.405  | 0.648 | 0.0952 | 0.336  |
>
> [1] Xiang et al. "Structured 3d latents for scalable and versatile 3d generation." CVPR 2025.

---

> > ### Author Rebuttal · Reviewer_92rs · 2026-04-03
> >
> > - The additional evidence on larger-scale evaluation, cross-model generalization, and robustness to imperfect masks is useful and makes the paper stronger.
> > - The rebuttal resolves a meaningful portion of my concerns and improves my confidence in the paper.
> >
> > Overall, I appreciate the authors’ effort in the rebuttal and will maintain my original score.

---

> > > ### Author Response · Authors · 2026-04-07
> > >
> > > Dear Reviewer 92rs,
> > >
> > > Thank you for your response and for acknowledging that a meaningful portion of your concerns has been resolved. We appreciate your time and the positive feedback on our rebuttal.
> > >
> > > Best regards,
> > > The Authors

---

### Official Review · Reviewer_VPF6 · 2026-03-01

**Soundness:** 2
**Presentation:** 2
**Significance:** 2
**Originality:** 3
**Overall Recommendation:** 3
**Confidence:** 4

**Summary:**

TIMI is a training-free framework for Image-to-3D multi-instance generation that disentangles spatial priors from pre-trained models. By introducing ISG and SGU, it achieves state-of-the-art layout accuracy and object distinctness without additional training costs.

**Compliance With Llm Reviewing Policy:**

Affirmed.

**Final Justification:**

While the authors expanded the evaluation set, validated generalization to Trellis, and added comparison with SAM 3D, critical concerns remain. The inconsistent evaluation results between this work and the original SAM 3D paper cast doubt on experimental reliability. These issues weaken the soundness of the comparison, so I maintain my original score.

**Key Questions For Authors:**

1. To provide a truly fair assessment, the authors should decouple the benefits of the proposed modules (ISG and SGU) from the backbone's inherent strength, perhaps by implementing the baselines on the same pre-trained model where possible.
2. The authors should expand the test set to ensure the results are representative and not subject to selection bias.
3. To demonstrate the generalizability and robustness of their method, the authors should conduct experiments using other prominent models, such as Trellis or Trellis2.
4. A qualitative and quantitative comparison with SAM 3D is essential for a complete literature review and performance benchmark.

**Limitations:**

yes

**Strengths And Weaknesses:**

Strengths
1. Training-free with an inference time of ~59s, significantly faster than DPA.
2. Achieves a SSR of 0.809, outperforming training-based MIDI (0.737).
3. ISG and SGU successfully mitigate instance entanglement in the early denoising stage.

Weaknesses
1. TIMI uses the powerful Hunyuan3D 2.0, whereas MIDI uses a likely smaller rectified flow architecture and DPA relies on the outdated Shap-E. The performance gap may stem from the backbone rather than the proposed method.
2. The study evaluates only ~70 samples total across three domains. This is statistically weak compared to baselines like MIDI, which used 1,000 samples.
3. As a training-free framework, the authors should prove generalizability by applying ISG/SGU to other models like Trellis, rather than relying exclusively on Hunyuan3D 2.0.
4. The manuscript overlooks "SAM 3D: 3Dfy Anything in Images," a key related work in composable 3D scene generation that warrants comparison.

---

> ### Author Rebuttal · Authors · 2026-03-31
>
> We sincerely thank the reviewer for the thoughtful and constructive comments. We hope the following clarifications can address your common concerns.
>
> **W1/Q1 : Backbone fairness**
>
> **Response:** We acknowledge that different methods are built upon different underlying models, and backbone strength can influence absolute performance. To decouple this factor, we have conducted comparisons on the same backbone in the paper. Specifically, compared to the original Hunyuan3D 2.0, TIMI achieves consistent improvements across all metrics in Tab. 1, demonstrating clear gains under an identical backbone. The ablation study in Tab. 2 further shows that performance improves progressively with ISG and SGU, confirming that each component contributes to the final performance gain.
>
> We also note that methods such as MIDI are built upon their own image-to-3D generation models, whose exact backbone configurations are not explicitly specified, making strict same-backbone comparisons difficult. Nevertheless, the controlled experiments on Hunyuan3D already provide strong evidence for the effectiveness of our method. Overall, these results demonstrate that ***the observed improvements are primarily attributed to our proposed modules (ISG and SGU), rather than the backbone itself.***
>
> **W2/Q2: Test Dataset size**
>
> **Response:** We agree that the original evaluation size may raise concerns about statistical robustness. To address this, we expand the test set to 500 samples and report the updated results in the table below. We observe that ***TIMI consistently outperforms both the base model and MIDI across all metrics in terms of global and local spatial fidelity.*** We further report confidence intervals for all metrics, which are relatively small, indicating that the improvements are stable and not due to sample noise. We note that, due to the substantially higher inference cost, scaling DPA to the same evaluation size is computationally prohibitive, and we therefore focus on methods with comparable evaluation settings. We will include the expanded evaluation results for all applicable methods in the revised version.
>
> | Method | LCD ↓  | CD_S ↓| FS_S ↑ | SSR ↑| CD_O ↓ | FS_O ↑|
> | --- | -- | -- | -- | -- | --- | --- |
> | Hunyuan3D 2.0 | 0.609 ± 0.023 | 0.0451 ± 0.0029 | 0.398 ± 0.018  | 0.695 ± 0.019  | 0.1062 ± 0.0071 | 0.268 ± 0.018  |
> | MIDI  | 0.608 ± 0.023  | 0.0494 ± 0.0031 | 0.324 ± 0.017     | 0.690 ± 0.018  | 0.1129 ± 0.0062   | 0.189 ± 0.014  |
> | TIMI  | **0.596 ± 0.023** | **0.0426 ± 0.0029** | **0.410 ± 0.018** | **0.705 ± 0.020** | **0.1024 ± 0.0069** | **0.274 ± 0.018** |
>
> **W3/Q3: Generalizability to other backbones**
>
> **Response:** We agree that evaluating TIMI on other backbones is important to demonstrate its generalizability and robustness. To this end, we further apply our method to another representative model, Trellis[1]. We observe consistent improvements across multiple metrics, including global and local spatial fidelity, demonstrating that TIMI effectively enhances both layout alignment and instance distinctiveness on Trellis. These results confirm that ***TIMI is not tied to a specific backbone and generalizes well across different I23D models.***
>
> | Method  | LCD ↓     | CD_S ↓     | FS_S ↑    | SSR ↑     | CD_O ↓     | FS_O ↑    |
> | --- | -- | ----- | ----- | ---- | ------ | ------ |
> | Trellis        | 0.692     | 0.0668     | 0.221     | 0.776     | 0.0899     | 0.131     |
> | Trellis + TIMI | **0.635** | **0.0622** | **0.273** | **0.785** | **0.0813** | **0.156** |
>
> **W4/Q4: Comparison with SAM 3D**
>
> **Response:** We thank the reviewer for pointing out SAM3D [2] as a relevant work. We have now included both qualitative and quantitative comparisons with SAM3D to provide a more complete evaluation. As shown in R-Fig.3 (***https://anonymous.4open.science/r/TIMI-rebuttal-7BD9***) and the updated quantitative results below, TIMI consistently outperforms SAM3D in terms of spatial fidelity, including both global layout alignment (LCD, CD-S, FS-S) and local instance distinctiveness (SSR, CD-O, FS-O). Overall, ***these results provide a more comprehensive benchmark and further validate the advantage of TIMI in multi-instance 3D generation.***
>
> | Method | LCD ↓ | CD-S ↓     | FS-S ↑ | SSR ↑ | CD-O ↓  | FS-O ↑   | Time ↓  |
> |- | - |- |-- |- |- |- |- |
> | Hunyuan3D 2.0   | 0.627     | 0.0492     | 0.450     | 0.697     | 0.0986     | 0.339     | **54.2s** |
> | DPA  | 0.649     | 0.0662     | 0.249     | 0.743     | 0.1620     | 0.124     | 783s      |
> | SAM3D  | 0.609     | 0.0595     | 0.262     | 0.740     | 0.1103     | 0.182     | 85.3s     |
> | MIDI    | 0.634     | **0.0409** | 0.396     | 0.737  | **0.0760** | 0.312     | 90.1s     |
> | **Ours (TIMI)** | **0.598** | _0.0424_   | **0.458** | **0.809** | _0.0855_   | **0.353** | _59.2s_   |
>
> [1] Xiang et al. "Structured 3d latents for scalable and versatile 3d generation." CVPR 2025.
>
> [2] Chen et al. "Sam 3d: 3dfy anything in images." arXiv:2511.16624.

---

> > ### Author Rebuttal · Reviewer_VPF6 · 2026-04-02
> >
> > Thank you for your thorough response and the supplementary experiments conducted to address my concerns. After carefully reviewing your revised analyses and results, I still hold some remaining questions and differing viewpoints, and thus I will maintain my original score.
> >
> > 1. Regarding backbone fairness: Although your TIMI framework improves the 3D scene reconstruction performance of Hunyuan3D 2.0, Hunyuan3D 2.0 is not originally designed for multi-instance 3D scene reconstruction tasks, making such comparisons less fair. Furthermore, the results of Trellis + TIMI are clearly inferior to MIDI, which strongly suggests that the backbone choice has a decisive impact on final performance.
> >
> > 2. Inconsistent comparison with SAM 3D: There is a noticeable contradiction in the evaluation against SAM 3D. From the supplementary materials at https://anonymous.4open.science/r/TIMI-rebuttal-7BD9/README.md, SAM 3D performs poorly and even appears ineffective in your experimental setting, with quantitative results worse than MIDI. However, in the original SAM 3D paper (https://arxiv.org/pdf/2511.16624), Figure 7 and Table 3 clearly show that SAM 3D achieves strong performance and outperforms MIDI. This conflicting outcome is confusing and casts doubt on the reliability of your comparative evaluation.

---

> > > ### Author Response · Authors · 2026-04-03
> > >
> > > We are glad that our previous response helped clarify some of the reviewer’s concerns, and we thank the reviewer for the opportunity to further improve our work. We address the remaining questions below.
> > >
> > > **A1. Regarding Backbone Fairness:**
> > >
> > > We agree that the capability of the backbone plays an important role in determining the final performance. It is important to emphasize that our method is a ***training-free framework***, which can be directly applied to different pre-trained I23D models without additional training or modification of the model architecture, with the goal of extending I23D models from single-object generation to multi-instance scenarios.
> > >
> > >    From the experimental results, we observe two key findings. First, on Hunyuan3D 2.0, our method significantly improves its multi-instance generation capability. Second, on a relatively weaker backbone (Trellis), TIMI still brings consistent performance gains, although the absolute performance remains below MIDI. These observations indicate that, while the backbone determines the upper bound of performance, ***TIMI consistently provides additional gains across different backbones, demonstrating that it is a general, training-free guidance approach that is decoupled from backbone design***.
> > >    Moreover, as a training-free method, TIMI can naturally benefit from future advances in I23D models, offering a scalable direction without requiring additional multi-instance training data.
> > >
> > >
> > > **A2. Regarding Comparison with SAM 3D:**
> > >
> > > We thank the reviewer for raising this important concern. We agree with the reviewer that SAM3D demonstrates strong performance in its original paper. To ensure a rigorous and faithful reproduction, we strictly follow the official implementation and verify that the generated results are reasonable. For fair comparison, we further convert the outputs of all methods into a unified mesh representation for both qualitative and quantitative evaluation. To eliminate potential discrepancies caused by implementation details, we additionally re-run SAM3D using its official multi-object inference pipeline, strictly following its Gaussian Splatting (GS) representation.
> > >
> > >    We provide the updated qualitative results in the anonymous link (R-Fig4, ***https://anonymous.4open.science/r/TIMI-rebuttal-7BD9/README.md***), ***which suggests that our reproduction is consistent with the official pipeline.*** Under the same input conditions and experimental settings, we observe that SAM3D’s qualitative performance shows certain differences compared to its original paper, and in some cases performs comparably to MIDI. We believe this discrepancy mainly arises from differences in task settings, such as input assumptions (high-quality segmentation vs. off-the-shelf segmentation) and variations in scene distributions.
> > >
> > >    For quantitative evaluation, we further clarify that SAM3D is originally designed with a Gaussian representation and does not provide a standard pipeline for multi-object mesh export. In our evaluation, all methods are assessed under a unified mesh-based protocol (e.g., CD, F-score). Converting GS representations to mesh may introduce inconsistencies across representations and affect geometry-sensitive metrics such as CD and F-score. Meanwhile, we observe that on ***metrics more related to layout alignment and instance separation*** (e.g., LCD and SSR), ***SAM3D still outperforms MIDI, which is consistent with its original conclusions to some extent.***
> > >
> > >    Overall, we make every effort to ***strictly follow the official implementation without altering the core model or algorithmic components***, and conduct comparisons under unified input and evaluation settings. We will also release the evaluation code to improve reproducibility and transparency. We hope these clarifications help address the reviewer’s concerns.

---

### Official Review · Reviewer_uzZw · 2026-03-03

**Soundness:** 3
**Presentation:** 3
**Significance:** 3
**Originality:** 3
**Overall Recommendation:** 4
**Confidence:** 2

**Summary:**

This paper proposes TIMI, a training-free framework for image-to-3D multi-instance generation. The authors discuss a significant problem: existing multi-instance 3D generation methods either suffer from error accumulation in compositional pipelines or substantial training overhead through fine-tuning. Observing that pre-trained I23D models already possess spatial priors underutilized due to instance entanglement, TIMI introduces two modules — Instance-aware Separation Guidance (ISG) and Spatial-stabilized Geometry-adaptive Update (SGU). Experiments on synthetic, real-world, and stylized datasets show improvements over baselines in layout accuracy and instance separation.

**Compliance With Llm Reviewing Policy:**

Affirmed.

**Final Justification:**

My concerns have been addressed, and I will keep the positive score.

**Key Questions For Authors:**

**Q1.** Has any robustness analysis been conducted regarding mask quality from Grounded-SAM? Understanding degradation under imperfect segmentation is critical for practical applicability.


**Q2.** How does TIMI perform as the number of instances increases (e.g., 5+) or when instances are severely occluded? Performance trends across varying instance counts would substantially strengthen the paper.


**Q3.** Can the authors validate TIMI on at least one other base model (e.g., Trellis) to demonstrate that the approach generalizes beyond Hunyuan3D 2.0's specific architecture and spatial priors?

**Limitations:**

Yes.

**Strengths And Weaknesses:**

## Strengths

**Well-motivated training-free paradigm.** The observation that pre-trained I23D models possess meaningful but entangled spatial priors provides a compelling motivation. The training-free approach yields clear practical advantages — ~59s inference versus ~90s (MIDI) and ~783s (DPA) — while eliminating training overhead entirely.

**Complementary and well-structured module design.** ISG and SGU address distinct failure modes: ISG targets instance entanglement through structure-aware spatial weighting that avoids degenerate solutions like attention collapse, while SGU prevents geometric degradation from raw gradient updates. Each design choice is technically well-motivated.

**Thorough ablation studies.** The paper systematically ablates all key components and hyperparameters, providing clear insight into each module's contribution and sensitivity.

---

## Weaknesses

**Inconsistent quantitative superiority.** MIDI outperforms TIMI on both CD-S and CD-O in Table 1. The authors justify this by arguing LCD better reflects layout alignment, but LCD is newly proposed here without validated correlation to human perception. Similarly, SSR only counts instance numbers without considering position or quality, making it an overly simplistic separation metric.

**Narrow baseline and model coverage.** Only three baselines are compared, and TIMI is validated on a single base model (Hunyuan3D 2.0). Without experiments on alternative foundations (e.g., Trellis) or comparisons with other recent methods mentioned in Related Works (e.g., ComboVerse), both generalizability and comparative strength remain unclear.

---

> ### Author Rebuttal · Authors · 2026-03-31
>
> We appreciate the reviewer for acknowledging the contributions of our work and all the constructive suggestions. We hope the following clarifications can address your concerns.
>
> **W1: Quantitative Superiority and Evaluation Metrics**
>
> **Response:** We clarify that different metrics capture different aspects of 3D generation quality. CD and F-Score primarily evaluate geometric accuracy, where TIMI achieves performance comparable to MIDI (e.g., higher F-Scores despite slightly worse CD). Our primary focus is spatial fidelity, including layout alignment and instance distinctiveness, which is not fully captured by CD. Although LCD is newly introduced, we find that its ranking (TIMI > Hunyuan3D > MIDI) is consistent with human preference (Fig. 6). Regarding SSR, it is designed to capture instance-level separation, while geometric quality is already evaluated by CD and F-Score. Overall, these metrics are complementary, together providing a more complete evaluation of multi-instance generation.
>
> **W2/Q3: Cross-model Generalization and  Baselines**
>
> **Response:** To evaluate generalization beyond a single backbone, we apply TIMI to another representative I23D model, Trellis [1], without modifying its architecture. As shown below, TIMI consistently improves both global and local spatial fidelity on Trellis. These results demonstrate that ***TIMI is not tied to a specific backbone and generalizes across different I23D models.***
> | Method | LCD ↓ | CD_S ↓ | FS_S ↑| SSR ↑ | CD_O ↓ | FS_O ↑ |
> | - | - | - | - | - | -- | - |
> | Trellis | 0.692 | 0.0668  | 0.221  | 0.776  | 0.0899  | 0.131|
> | Trellis + TIMI | **0.635** | **0.0622** | **0.273** | **0.785** | **0.0813** | **0.156** |
>
> Regarding baseline coverage, ***since ComboVerse [2] is not publicly available***, we instead include SAM3D [3] as an additional baseline. TIMI achieves better performance in both qualitative results (R-Fig.3, ***https://anonymous.4open.science/r/TIMI-rebuttal-7BD9***) and quantitative metrics, further supporting the strength of our approach.
>
> | Method | LCD ↓  | CD-S ↓  | FS-S ↑    | SSR ↑  | CD-O ↓| FS-O ↑ | Time ↓ |
> |- |- | - |- |-- |- | - |- |
> | Hunyuan3D 2.0   | 0.627| 0.0492   | 0.450  | 0.697 | 0.0986 | 0.339 | **54.2s** |
> | DPA    | 0.649     | 0.0662 | 0.249  | 0.743  | 0.1620 | 0.124  | 783s  |
> | SAM3D  | 0.609  | 0.0595  | 0.262 | 0.740  | 0.1103  | 0.182 | 85.3s  |
> | MIDI  | 0.634 | **0.0409** | 0.396  | 0.737 | **0.0760** | 0.312  | 90.1s   |
> | **Ours (TIMI)** | **0.598** | *0.0424*   | **0.458** | **0.809** | *0.0855*   | **0.353** | *59.2s*   |
>
> **Q1: Mask Robustness**
>
> **Response:** We thank the reviewer for raising the important question of robustness to mask quality. In practice, TIMI is robust to mask imperfections, as observed in qualitative results with occlusion and overlapping objects (Fig.1, 3, 4, 5).  We further visualize the corresponding masks for these cases, available at ***https://anonymous.4open.science/r/TIMI-rebuttal-7BD9***. To systematically analyze this, we conduct controlled experiments by degrading the input masks using various perturbations, including boundary erosion/dilation, random region dropout, and missing instances. As shown in the table below, ***TIMI degrades gracefully under different types of mask corruption, with only limited impact on spatial fidelity. This indicates that TIMI remains stable under imperfect segmentation and does not rely on precise masks.***
>
> | Mask Setting | LCD ↓ | CD-S ↓ | FS-S ↑ | SSR ↑ | CD-O ↓ | FS-O ↑ |
> | - | - | - | - | - | - | - |
> | Clean| 0.551 | 0.0468 | 0.428  | 0.672 | 0.0894 | 0.336  |
> | Boundary Noise  | 0.559 | 0.0553 | 0.414  | 0.728 | 0.1087 | 0.315  |
> | Dropout | 0.630 | 0.0525 | 0.408  | 0.662 | 0.1032 | 0.308  |
> | Missing Instance | 0.572 | 0.0521 | 0.405  | 0.648 | 0.0952 | 0.336  |
>
> **Q2: Increasing Instance Count and Severe Occlusion**
>
> **Response:** TIMI maintains stable performance as the number of instances increases and does not exhibit performance collapse in more complex scenes. As shown in R-Fig.2 (***https://anonymous.4open.science/r/TIMI-rebuttal-7BD9***), when the number of instances increases (e.g., 5+), TIMI consistently outperforms the base model across all metrics (CD-S, FS-S, CD-O, FS-O). Importantly, the performance remains stable even at higher instance counts (e.g., 7–9), indicating that ***TIMI scales well to complex multi-instance scenarios.*** Severe occlusion is illustrated in Q1 qualitative results (R-Fig. 1). For example, in cases where a nightstand is almost entirely occluded by a bed, with only a small visible region, TIMI can still generate it as a distinct instance without merging it with nearby objects, while preserving the correct spatial layout.
>
> [1] Xiang et al. "Structured 3d latents for scalable and versatile 3d generation." CVPR 2025.
>
> [2] Chen et al. "Comboverse: Compositional 3d assets creation using spatially-aware diffusion guidance." ECCV 2024.
>
> [3] Chen et al. "Sam 3d: 3dfy anything in images." arXiv:2511.16624.

---

> > ### Author Rebuttal · Reviewer_uzZw · 2026-04-04
> >
> > Thank you for the detailed rebuttal. My concerns have been addressed, and I will keep the positive score.

---

> > > ### Author Response · Authors · 2026-04-07
> > >
> > > Dear Reviewer uzZw,
> > >
> > > Thank you for your response and for confirming that your concerns have been fully addressed. We sincerely appreciate your constructive comments throughout the review process, which have helped enhance the clarity of our work. We are grateful for your support.
> > >
> > > Best regards,
> > > The Authors

---

### Official Review · Reviewer_cosH · 2026-03-11

**Soundness:** 3
**Presentation:** 3
**Significance:** 3
**Originality:** 2
**Overall Recommendation:** 4
**Confidence:** 4

**Summary:**

This paper proposes a framework called TIMI, a high-precision image-to-3D multi-instance generation framework that requires no training. The authors observe that while pre-trained I23D models possess meaningful spatial priors, they suffer from instance entanglement. To address this, they introduce two key modules: Instance-Aware Separation Guidance (ISG), which suppresses instance entanglement by focusing on separation loss during anchoring and early denoising stages; and Spatial Stable Geometric Adaptive Update (SGU), which provides stable guidance through Gaussian smoothing and adaptive gradient modulation. Experiments on synthetic, real-world, and stylized datasets demonstrate that the framework fails to achieve high-quality generation and improve inference efficiency.

**Compliance With Llm Reviewing Policy:**

Affirmed.

**Final Justification:**

The author has clearly answered my question, so I've decided to raise my rating.

**Key Questions For Authors:**

1.Since TIMI guides the generation of multiple instances, will this guidance conflict with the multi-view consistency of the underlying I23D model, especially at extreme camera angles?
2.Since TIMI relies on Grounded-SAM for mask extraction, how sensitive is the entire process to mask quality? Specifically, in cases of severe occlusion or very fine boundaries, will a decrease in mask quality lead to 3D generation failure? Experiments controlling mask quality can enhance the demonstration of the process's robustness.
3.The smoothing parameter σ in the SGU module has a significant impact on the results. In this paper, σ=1.5 was chosen as the default value, but different types of objects may have different smoothing requirements. Are there any experiments demonstrating the universality of this fixed parameter across different scenarios?

**Limitations:**

yes

**Strengths And Weaknesses:**

Strengths ：
1.The core motivation is reasonable: utilizing the underutilized spatial prior information in pre-trained I23D models is a meaningful and computationally efficient alternative in practical applications, which can replace fine-tuning-based methods.
2.The combination of ISG and SGU is technically compatible. The geometrically adaptive modulation in SGU cleverly solves the problem of heterogeneous sensitivity between different geometries.
3.The inference speed (~ 59s) is significantly lower than DPA (783s) and competitive with the base model, which is a meaningful practical advantage.
Weaknesses:
1.Inference Cost Analysis: While the "Guidance" steps (ISG/SGU) save training time, they increase computational overhead with each denoising iteration. This paper lacks a detailed comparison of actual runtime with the base model during the inference phase.
2.This paper does not adequately discuss how the method handles “nested” instances or objects with significant overlap in a two-dimensional input where spatial masks are difficult to separate.
3.The Separation Success Rate (SSR) metric only counts instance numbers without considering spatial overlap or geometric quality, which may not faithfully reflect true instance separation. Two heavily overlapping instances would still yield SSR=1.0 if the count matches.

---

> ### Author Rebuttal · Authors · 2026-03-31
>
> We appreciate the reviewer’s insightful comments and suggestions. We hope the following clarifications help address the concerns.
>
> **W1: Inference Cost Analysis**
>
> **Response:**  ***We have reported the end-to-end inference time in Tab.1, where TIMI takes 59.2s vs. 54.2s for Hunyuan3D (\~5s overhead), indicating that it remains efficient in the full pipeline.*** To further clarify, we provide a detailed analysis focusing on the denoising stage (50 steps). Under this setting, runtime increases from 17.52s to 22.38s, mainly due to ISG (3.93s), while SGU is negligible (0.01s).  Importantly, ***this inference overhead is localized and well-controlled,*** as guidance is applied only to early steps and partial layers. Overall, TIMI achieves a favorable trade-off between efficiency and multi-instance spatial fidelity.
>
> |Method|Base Denoising(s)|ISG(s)|SGU(s)|Total(s)|
> |-|-|-|-|-|
> |Hunyuan3D 2.0|17.52|-|-|17.52|
> |Hunyuan3D 2.0 + TIMI|18.44|3.93|0.01|22.38|
>
> **W2: Nested / overlapping instances**
>
> **Response:** We thank the reviewer for raising the concern regarding nested or heavily overlapping instances, where 2D masks are difficult to separate. In practice, ***such cases (e.g., due to occlusion) are already included in our paper,*** as shown in
> qualitative examples (Fig. 1, 3, 4, 5).  Inspired by your suggestion, we provide additional visual cases along with their corresponding masks in R-Fig.1 (***https://anonymous.4open.science/r/TIMI-rebuttal-7BD9***). These masks may exhibit overlap, nesting, or missing regions, yet remain sufficient to guide generation, indicating that TIMI does not rely on precise instance segmentation. Overall, while extremely ambiguous cases may remain challenging, our results demonstrate that ***TIMI is robust to overlapping instances in practical scenarios.***
>
> **W3: SSR metric**
>
> **Response:** We would like to clarify that ***SSR is computed in 3D space rather than based on 2D projections,*** and therefore is not a simple count of instances in 2D. Specifically, even if two objects appear overlapping from certain views in 2D, as long as they remain geometrically separate in 3D, they are treated as distinct instances in the ground truth. Conversely, if the generated result produces fused geometry, it will be counted as a single instance, resulting in a lower SSR. Thus, ***the failure case described by the reviewer—where heavily overlapping instances still yield a high SSR—does not occur under our 3D-based evaluation.*** Importantly, SSR is used together with geometry-aware metrics (e.g., CD, F-score), ensuring that both instance separation and geometric quality are properly evaluated.
>
> **Q1: Multi-view consistency**
>
> **Response:** TIMI does not conflict with the underlying multi-view consistency of the I23D model. It operates as a guidance mechanism on a frozen model and does not alter its inherent 3D generation capability. To directly verify this, we render the generated results from multiple viewpoints (including extreme angles) using existing outputs, as shown in the R-Fig.1 (***https://anonymous.4open.science/r/TIMI-rebuttal-7BD9***). The geometry remains consistent and stable across views, indicating that ***TIMI preserves multi-view consistency while improving spatial fidelity.***
>
> **Q2: Sensitivity to mask quality**
>
> **Response:** To address the concern on mask quality, we analyze the sensitivity of TIMI and find that it remains robust to common mask imperfections, without requiring perfectly accurate masks. ***It produces spatially coherent 3D outputs with consistent layouts and well-separated instances even with imperfect Grounded-SAM masks (see R-Fig.1, https://anonymous.4open.science/r/TIMI-rebuttal-7BD9).*** We further conduct controlled perturbations (boundary noise, dropout, and missing instances). As shown in the table below, the performance degrades gracefully under moderate mask corruption, with limited impact on spatial fidelity. **Overall, both qualitative and quantitative results demonstrate that TIMI does not require precise segmentation and can reliably preserve spatial fidelity from coarse masks.**
>
> |Mask Setting|LCD↓|CD-S↓|FS-S↑|SSR↑|CD-O↓|FS-O↑|
> |-|-|-|-|-|-|-|
> |Clean|0.551|0.0468|0.428|0.672|0.0894|0.336|
> |Boundary Noise|0.559|0.0553|0.414|0.728|0.1087|0.315|
> |Dropout|0.630|0.0525|0.408|0.662|0.1032|0.308|
> |Missing Instance|0.572|0.0521|0.405|0.648|0.0952|0.336|
>
> **Q3: Universality of σ**
>
> **Response:** We agree that σ affects performance, as reflected in our ablation study in Tab.7. At the same time, Tab.7 shows that performance changes smoothly with σ, and σ=1.5 provides a balanced trade-off, indicating a stable operating range rather than a narrowly tuned optimum. We also note that we use the same σ across all datasets, consistently producing stable and well-separated results without per-scene tuning. While different scenes may have slightly different optimal values, σ=1.5 serves as a robust and practical default.

---

> > ### Author Rebuttal · Reviewer_cosH · 2026-04-07
> >
> > Thank you for the author's reply. My problem has been resolved, and I have decided to increase my review score.

---

> > > ### Author Response · Authors · 2026-04-08
> > >
> > > Dear Reviewer cosH,
> > >
> > > Thank you for your time and constructive feedback, and for the increase in your score. We are glad that our rebuttal has addressed your concerns.
> > >
> > >
> > > Best regards,
> > > The Authors

---

### Decision · Program_Chairs · 2026-04-30

**Decision:**

Accept (regular)

**Comment:**

The paper proposes a training free method to improve the spatial fidelity
of multi-instance generation for existing image-to-3D models. It received
3 weak accept and 1 weak reject. The concerns from the 3 positive reviewers
are well addressed during the rebuttal.
The opinions from the negative reviewer mainly focus on the unfair comparisons
against MIDI due to different backbones, insufficient number of evaluation samples,
applying the method to more models, and comparisons against SAM3D.
The authors make replies to all these questions. The reviewer asks follow-up
questions on backbone fairness and comparison against SAM3D. From the authors
rebuttal, I believe the authors have demonstrated the effectiveness of the proposed
method on different backbones, and have made their efforts to make fair comparisons
against MIDI and SAM3D. Therefore, the AC believes that the reviewers' concerns are
well addressed and recommends acceptance of the paper.